# In-Vitro Inactivation of Sabin-Polioviruses for Development of Safe and Effective Polio Vaccine

**DOI:** 10.3390/vaccines8040601

**Published:** 2020-10-13

**Authors:** Asmaa A. Abd-Elghaffar, Mohamed E. Rashed, Amal E. Ali, Magdy A. Amin

**Affiliations:** 1Biotechnology Unit, National Organization for Research and Control of Biological (NORCB), Giza 12654, Egypt; ph.asmaa.ahmed@gmail.com; 2Bacterial Control Unit, National Organization for Research and Control of Biological (NORCB), Giza 12654, Egypt; mohamed_elsayedeg@yahoo.com; 3Microbiology and Immunology Department, Faculty of Pharmaceutical Sciences & Pharmaceutical Industries, Future University in Egypt, New Cairo 12311, Egypt; 4Department of Microbiology and Immunology, Faculty of Pharmacy, Cairo University, Cairo 11562, Egypt; magdy.amin@pharma.cu.edu.eg

**Keywords:** polio, inactivation, viral vaccine, immunization, developing countries

## Abstract

After years of global collaboration; we are steps away from a polio-free world. However, the currently conventional inactivated polio vaccine (cIPV) is suboptimal for the post eradication era. cIPV production cost and biosafety hazards hinder its availability and coverage of the global demands. Production of IPV from the attenuated Sabin strains (sIPV) was an ideal solution and scientists work extensively to perfect a safe, effective and affordable sIPV. This study investigated the ability of hydrogen peroxide (H_2_O_2_), ascorbic acid (AA) and epigallocatechin-3-gallate (EGCG) as alternatives for Formaldehyde (HCHO) to inactivate Sabin-polioviruses strains for sIPV production. Sabin-polioviruses vaccine strains were individually treated with AA, EGCG or H_2_O_2_ and were compared to HCHO. This was investigated by determination of the inactivation kinetics on HEP_2C_ cells, testing of D-antigen preservation by ELISA and the immune response in Wistar rats of the four vaccine preparations. H_2_O_2_, AA and EGCG were able to inactivate polioviruses within 24 h while HCHO required 96 h. Significant high D-antigen levels were observed using AA, EGCG and H_2_O_2_ compared to HCHO. Rat sera tested for neutralizing antibodies showed comparable results. These findings support the idea of using these inactivating agents as safe and time- saving alternatives for HCHO to produce sIPV.

## 1. Introduction

Poliomyelitis is a serious disabling highly contagious viral infection caused by polioviruses. In 1988, The Global Polio Eradication Initiative (GPEI) was founded to achieve global eradication of polio and since then, different strategies were adopted for that goal. The continuous efforts and the global collaboration reduced the number of cases dramatically via massive immunization campaigns with oral polio vaccine (OPV) which was a perfect choice due to its affordability and superior mucosal immunity [1]. However, OPV is no longer a favorable option for the risk of generating vaccine-derived polioviruses (VDPV) and vaccine-associated paralytic poliomyelitis (VAPP) cases. The polio end-game plans developed by GPEI (2013–2018 and 2019–2023) included the global gradual withdrawal of OPV starting with type 2 and introduction of one dose at least of inactivated polio vaccine (IPV) into the routine immunization program [2]. Also, wild poliovirus type 3 (WPV3) was last detected in 2016 and currently is certified as eradicated [3]. This will subsequently lead to the cessation of OPV type 3 from the global immunization and emphasizes the importance of affordable IPV production [4,5].

Various strategies were investigated to reduce IPV cost including reduction of either the number or the volume of administered vaccine doses via intradermal administration and this approach was implemented in immunization campaigns of certain countries to overcome the IPV shortage [6,7]. Reduction of the antigen content and boosting the immune response through using adjuvants is another promising approach which is widely investigated [8,9,10]. Finally, Optimization of the vaccine manufacturing process via the usage of less infectious virus strains became a priority to allow IPV production in middle- and low-income countries due to the unacceptable biosafety risks accompanied with conventional IPV production. IPV developed from live-attenuated Sabin strains (sIPV) was a reasonable alternative and many studies proved its efficacy and affordability [11,12,13,14]. In 2012, manufacturers in Japan and China were licensed for production and marketing of Sabin-IPV and this vaccine is used there extensively.

Although formaldehyde is the classic inactivating agent for polio vaccine from 1954 till now, formaldehyde has many drawbacks. It is proved that using formaldehyde adversely affects the antigenic structure of a number of viruses during vaccines production [15]. Traditional poliovirus inactivation process requires about 2 to 3 weeks at 37 °C to be completely inactivated. This is a time-consuming process besides the thermal degradation and destruction of important epitopes due to virus exposure to elevated temperatures for a prolonged time. Therefore, it is essential to find safe and affordable other inactivating agents.

Hydrogen peroxide (H_2_O_2_) is an oxidizing agent which is used widely and safely in different disinfection and sterilization processes [16] and was investigated previously as an inactivating agent for different vaccines [17,18,19]. The fact that H_2_O_2_ decomposes spontaneously into oxygen and water makes it a reasonable choice as it will not require complex purification steps during production. Ascorbic acid (AA) is a water-soluble vitamin. When catalyzed with cupric ions; it showed encouraging results as a potential inactivating agent for rabies vaccine production [20,21]. Epigallocatechin-3-gallate (EGCG) is the most notable component in green tea extract; previous studies proved its efficacy against a wide range of viruses and it was investigated as a novel inactivating agent for viral vaccines [22]. Taking that into consideration, this study investigated the aptitude of H_2_O_2_, AA and EGCG in causing complete and irreversible inactivation of Sabin polioviruses and their effect on antigenicity and immunogenicity of these viruses.

## 2. Materials and Methods

### 2.1. Virus Strains

Live attenuated Sabin polioviruses vaccine strains Type 1 (LSc, 2ab), Type 2 (P 712, Ch, 2ab) and Type 3 (Leon 12a, 1b) concentrated purified monovalent bulks were produced under cGMP conditions by a WHO pre-qualified manufacturer for oral polio vaccine (GSK, Wavre, Belgium) in MRC5 human diploid cells and kindly supplied from Cell Culture Department and Regional Laboratory for Enteric Viruses’ Diagnosis Center, VACSERA-Egypt. Viruses were aliquoted and stored at −80 °C. The supplied monovalent bulks were tested in GSK as per WHO recommendations including neurovirulence testing in transgenic mice, bacterial and fungal sterility and other relative tests [23].

### 2.2. Antibodies

Polyclonal antibodies against poliomyelitis viruses (types 1, 2 and 3) produced by Sanofi Pasteur (France) and were obtained from heifer after immunization with inactivated Salk strains Mahoney, MEF-1 and Saukett respectively. These antibodies are produced specifically to be used as coating antibodies in routine polio D- antigen ELISA assays.

Anti- D monoclonal antibodies (types 1, 2 and 3) were produced in rabbits by Sanofi Pasteur (France). The immunization strains used for preparation of these antibodies were inactivated Salk strains Mahoney, MEF-1 and Saukett respectively. They are used specifically as detection antibodies in routine polio D- antigen ELISA assays.

### 2.3. Animals and Cell-Line

Swiss albino mice (17–22 g, total n = 40), guinea-pigs (250–350 g, total n = 16) and male Wistar rats (30–40 g, total n = 25) were purchased from the animal facility of the holding company for biological products and vaccines (VACSERA) (Helwan, Egypt) and housed in laboratory animal facility, NORCB (Giza, Egypt). Housing of the animals was performed in compliance with standard laboratory conditions with access to food and water and in an environmentally controlled room with 12 h light/dark cycles. Hep-2c Cells were kindly supplied from cell culture laboratory of VACSERA (Giza, Egypt) and the cell line was preserved, retrieved, sub-cultured and maintained in biotechnology laboratory, NORCB. Animals and cell-line handling and safety precautions were performed according to guidelines [24,25]. Permission to conduct the study was obtained from Ethics Committee of Faculty of Pharmacy, Cairo University, Egypt (protocol code: MI1698-31/5/2016).

### 2.4. Determination of the Infectivity Titers of Polioviruses

Poliovirus serotypes were titrated using standard tissue culture technique where each serotype was 10 fold serially diluted in maintenance medium (Eagle’s Minimum Essential Medium with Earl’s balanced salt solution (EMEM with EBSS) (LONZA, Walkersville, Maryland, USA) + 2% fetal bovine serum (FBS) (Biowest, Nuaille, France, South America origin) and inoculated in 96 well tissue culture plates (Corning, NY, USA) with Hep-2c Cells. Plates were sealed using sterile sealer (Greiner, Frickenhausen, Germany) and incubated for 7 days with daily examination microscopically for the presence of cytopathogenic effect (CPE). Virus infectivity titers in log_10_CCID_50_/_mL_ were calculated by Kärber formula [26].

### 2.5. Viruses’ Inactivation Studies

#### 2.5.1. Inactivation Kinetics of Polioviruses Type 1, 2 and 3 Using Different Inactivating Agents

Prior inactivation, all monovalent bulks were filtered via 0.2 µm sterile syringe filters (Sartorius, Gottingen, Germany). PH before inactivation process was found to be 7.2 and at the end of inactivation was lowered by not more than 0.3 units. Different samples were taken from each monovalent bulk and treated individually with one of the experimental inactivating agents. Formaldehyde was used as the standard preparation to investigate the effect of other experimental inactivating agents compared to the effect of traditional HCHO. Aliquots from each treated monovalent bulk were withdrawn at different time intervals and were titrated for viral infectivity using Hep-2c cells in a tissue culture microtiter plate system [26].

Hydrogen peroxide (H_2_O_2_): the three PV serotypes were individually subjected to H_2_O_2_ (Carl Roth, Karlsruhe, Gremany) used at a final concentration of 3% in the virus suspension, kept at 2–8 °C and monitored over time [17]. An aliquot was withdrawn from each serotype at certain time intervals over 24 h and treated twice with catalase (MP Biomedical, Solon, Ohio, USA) at a final concentration of 12.5 U/mL for 10 min at room temperature to remove excess H_2_O_2_ [27].

Ascorbic acid (AA): each monovalent bulk was treated with ascorbic acid (Sigma, St. Louis, MO, USA) at a final concentration of 0.5 mg/mL and copper sulfate (Sigma, St. Louis, MO, USA) at 5 µg/mL [20]. Samples were kept at 2–8 °C aliquots were withdrawn over 24 h for analysis.

Epigallocatechin-3-gallate (EGCG): the viruses were treated with 0.5 mg/mL EGCG (Acros Organics, Geel, Belgium ) and incubated at 36 ± 1 °C [22]. Aliquots were withdrawn over 24 h for analysis.

Formaldehyde (HCHO): 0.003M HCHO (Sigma, St. Louis, MO, USA) was used on each virus type. Samples were incubated at 36 ± 1 °C [15]. The excess formaldehyde in the samples was neutralized via the addition of 0.25% sodium bisulfite (Acros Organics, Geel, Belgium). Aliquots were withdrawn daily for five days.

Each aliquot was titrated twice and results were represented as averages of the virus infectivity titers in log_10_CCID_50_/_mL_.

#### 2.5.2. Effectiveness of Inactivation

HCHO, H_2_O_2_, AA and EGCG inactivated viruses suspensions were tested using cell culture technique to detect residual infectious surviving viruses after inactivation in order to assure the efficiency of the inactivation process. Hep-2c cells were cultured in small tissue culture flasks (25 cm^2^) (SPL, Gyeonggi-do, Korea) and observed till formation of confluent sheets. Each inactivated serotype was inoculated into two flasks (each flask contained 3 mL sample and 6 mL nutrient medium with a ratio of 1:3). The flasks were incubated at 37 ± 1 °C for 10 days during this period; culture media were changed every 5 days. At the end of incubation period; 2 subcultures were made from each original flask and incubated for further 10 days. Cultures were examined microscopically for the presence of CPE [28,29].

### 2.6. Retention of Antigenicity after Inactivation

Inactivated monovalent samples were tested by an indirect ELISA method [30] to determine the ability of the different inactivating agents to inactivate Sabin-polioviruses and preserve the viruses’ antigenic moieties (D-antigens). Samples tested were HCHO, H_2_O_2_, AA and EGCG inactivated viruses’ suspensions as well as two control groups of untreated viruses one was kept at 2–8 °C and the other at 36 ± 1 °C. The assay was carried out using 96-wells ELISA plates, flat bottom (Greiner, Frickenhausen, Germany) which were coated with 100 µL/well anti-polio antibodies (PV1, PV2 or PV3) produced in heifer (Sanofi Pasteur, France). Plates were kept overnight at 2–8 °C. The next day, plates were washed 5 times using wash solution (PBS (Biobasic, Markham, Ontario, Canada), 0.025% tween 20 (Loba chemie, Mumbai, Maharashtra, India), 2% Milk (Bio-Rad, Irvine, CA, USA) as 200 µL/well with 1 min intervals to remove unbound antigens. The last washing solution is left in the plates for saturation and plates were kept at room temperature for at least 2 h. After that, plates were decanted and samples were two- fold serially diluted, added as 100 µL/well and incubated at 37 °C for 3 h. At the end of incubation period, plates were washed and 100 µL/well anti-polio antibodies (PV1, PV2 or PV3) produced in rabbit (Sanofi Pasteur, Val-de-Reuil, Normandy, France) was added to the plates. Plates were incubated for 2 h at 37 °C then transferred to 2–8 °C for overnight incubation. Next day, plates were washed and then anti-Rabbit IgG peroxidase conjugate produced in goat (Sigma, St. Louis, MO, USA), diluted 1:40,000 was added as 100 µL/well and incubated at 37 °C for 2 h, after that wash step was repeated then Tetramethylbenzidine (TMB) solution (KPL, Milford, MA, USA) was added as 100 µL/well and plates were kept at room temperature in dark place for 30 min. TMB Stopping solution (KPL, Milford, MA, USA) was added as 100 µL/well. The absorbance was read by TriStar 2 Multimode reader (Berthold Technologies, Bad Wildbad, Germany) at 450/620 nm wave length. Results were represented as mean absorbance of duplicate wells.

### 2.7. Trivalent Vaccine Preparation

HCHO-inactivation process was conducted for 10 days while viruses inactivated with H_2_O_2_, AA and EGCG were exposed to the inactivating agents for at least 24 h. Suspensions of either HCHO, H_2_O_2_, AA or EGCG- inactivated PV serotypes were mixed to contain 10^6.2^ logCCID_50_/_mL_ of PV1, 10^5.7^ logCCID_50_/_mL_ of PV2 and 10^6.3^ log CCID_50_/_mL_ of PV3 [31]. The trivalent bulks were filtered with 0.2 µm sterile filters (Sartorius, Germany) and stored at 2–8 °C.

### 2.8. Abnormal Toxicity Test

Test was performed according to European Pharmacopoeia 8.0 monograph 01/2008: 20,609 for human vaccines. For each trivalent preparation, 2 groups of Swiss mice weighing 17–22 g, (5 mice/group) in addition to 2 groups of healthy guinea-pigs weighing 250–350 g (2 guinea pigs/group) were inoculated via intra-peritoneal (IP) route at a dose of 0.5 mL/animal. All the inoculated animals were observed daily for any signs of illness during 7-days observation period.

### 2.9. Induction of Antibodies in Wistar Rats

HCHO, H_2_O_2_, AA and EGCG- inactivated trivalent bulks were inoculated into different groups of Wistar rats weighing 30–40 gm. (5 rats/group). Each rat received a single dose of 0.5 mL of each preparation intra-muscularly (IM) into the back of the thigh (0.25 mL/leg) [32]. A negative control group (5 rats) was injected with water for injection (WFI, Otsuka Pharm, Egypt). Blood was collected via retro-orbital route 21-days post immunization. Sera were separated, inactivated at 56 °C for 30 min. After that, sera from each group of rats were pooled together, aliquoted and stored at −80 °C till testing for IgG levels by ELISA and neutralizing antibodies by tissue culture technique [11].

### 2.10. Evaluation of Immune Response by ELISA

Rat sera were tested for the presence of polioviruses IgG antibodies by ELISA technique [30]. ELISA plates (Grenier, Dutch, Germany) were coated with 100 µL/well Sabin polioviruses monovalent bulks (GSK, Brentford, UK) types 1, 2 or 3 pre-diluted 1/5 in carbonate-bicarbonate buffer (Sigma Aldrich, St. Louis, MO, USA). Plates were kept overnight at 2–8 °C. The next day, plates were washed and saturated as described previously. After that, plates were decanted and sera samples were two-fold serially diluted in phosphate buffer saline (PBS) (Biobasic, Toronto, ON, Canada) +2%Bovine serum albumin (BSA) (Sigma, St. Louis, MO, USA) and dispensed as 100 µL/well. Plates were incubated for 2 h at 37 ± 1 °C then washed. Anti-Mouse IgG (whole molecule)-Peroxidase produced in rabbit, IgG fraction of antiserum (Sigma, St. Louis, MO, USA), diluted 1:40,000 as recommended in product information sheet was added as 100 µL/well and incubated at 37 °C for 2 h, after that a wash step was repeated then TMB (KPL, Kalamazoo, MI, USA) for 30 min followed by TMB Stopping solution (KPL, Kalamazoo, MI, USA). The absorbance was read by Multimode reader (TriStar2, Panzer, Germany) at 450/620 nm wave length and results were represented as mean absorbance of duplicate wells for each dilution. Cut-off value was considered as the average optical density of sample diluent plus two standard deviations (SD) [33]

### 2.11. Polioviruses-Neutralizing Antibodies Test

This test measures neutralizing antibody titers to poliovirus types 1, 2, and 3 produced in rat sera. The test principle is that the anti-poliovirus antibodies in a serum sample are allowed to be incubated with an infectious dose of polioviruses. The specific antibodies will bind to the virus and prevent infection of susceptible cells. Excess un-neutralized virus will infect the cells causing CPE.

Rat-sera samples were tested by a standard micro-neutralization procedure as per World Health Organization (WHO) guidelines [34]. Briefly, sera were initially diluted 1:4 in maintenance medium and added to row A of 96- wells tissue culture plates (Nunc, Rochester, NY, USA) from column 1 to 10 as 100 µL/well while columns 11 and 12 were kept as cell control which contained 100 µL/well of maintenance medium. Rows from B to H were filled with 50 µL/well of maintenance medium then serial two-fold dilutions of the tested sera were prepared by mixing and transferring 50 µL/well from A to B using multichannel micropipette (Eppendorf, Hamburg, Germany) till row H in the plates (dilutions from 1:4 to 1:512). Each group of rat sera was diluted in 3 plates to be tested against the 3 polioviruses. After that, Sabine polioviruses monovalent bulks (GSK, Belgium) types 1, 2 and 3 were diluted in maintenance medium to reach a concentration of 100TCID_50_/0.05 mL (range 50–200 TCID_50_/0.05 mL). A volume of 50 µL/well of each diluted poliovirus was added to all wells of their corresponding plates. Back titration plates were prepared containing three serial 10-fold dilutions of each prepared diluted virus type starting from 100TCID50/0.05mL. All Plates were sealed and incubated 3 h at 36 °C then overnight at 2–8 °C. Next day, Hep-2c Cell suspension was prepared and added as 100µL/well into all wells of all plates; plates were sealed and incubated at 37 ± 1 °C for 5–7 days. Plates were examined daily for detection of CPE. Log 50% neutralization titers were calculated by Kärber formula and the challenge viruses’ titers were also calculated. Antibody titers are expressed as reciprocals. The test was performed in triplicate for each serotype and the geometric mean titers (GMT) were compared.

### 2.12. Statistical Analysis

Data was processed using Graph Pad Prism 7 (Graph-Pad software Inc., San Diego, CA, USA). Groups were compared using unpaired parametric *t*-test. *p*-values less than 0.05 (typically ≤ 0.05) were considered statistically significant.

## 3. Results

### 3.1. Infectivity Titers of Polioviruses

Virus serotypes PV1, PV2 and PV3 were found to have the titers of 10^6.2^, 10^5.7^ and 10^6.3^ log CCID_50_/_mL_ respectively.

### 3.2. Kinetics of Inactivation

HCHO, H_2_O_2_, AA and EGCG showed their ability to inactivate the three serotypes in spite of the variation in their inactivation kinetics. HCHO-inactivated serotypes showed no CPE after 96 h of exposure. Our experimental inactivating agents were able to inactivate the same polioviruses’ suspensions in a significantly shorter exposure time. CPE was no longer detectable after 6 h in case of using H_2_O_2_. Ascorbic Acid and EGCG showed slower rates of inactivation as both AA and EGCG required more than 18 h to inactivate polioviruses. (Figure 1)

### 3.3. Effectiveness of Inactivation

To assure the total inactivation of polioviruses, HCHO-inactivation process was conducted for 10 days while viruses inactivated with H_2_O_2_, AA and EGCG were exposed to the inactivating agents for 24 h. All cultured and sub-cultured flasks showed no cytopathic effect during their incubation periods.

### 3.4. Retention of Antigenicity after Inactivation

The retention of D-antigens after inactivation as tested using ELISA technique is showed in (Figure 2). Results showed no significant difference between untreated viruses stored at 2–8 °C and the others kept at 36 ± 1 °C (*p* = 0.5131, 0.4900 and 0.1251 for PV1, PV2 and PV3 respectively). Concerning our experimental inactivating agents; antigenicity retained after AA-inactivation was significantly higher than HCHO for the three types (*p* = 0.0003, 0.0002 and 0.0032 respectively). Similarly, EGCG showed significantly higher OD values than HCHO (*p* = 0.0027, 0.0015 and 0.0001 respectively). H_2_O_2_ recorded significantly higher results for PV2 and PV3 (*p* = 0.0096 and 0.0068).Although the observed results of H_2_O_2-_ inactivated PV1 are higher than HCHO results, this increase was insignificant statistically (*p* = 0.5131).

### 3.5. Abnormal Toxicity

All animals survived till the end of observation period. There were no significant changes in their weight, food consumption or general activity and they showed no signs of illness.

### 3.6. Evaluation of Immune Response by ELISA

Anti-polio immunoglobulin’s for the three types were tested in the sera collected from rats. The OD responses obtained from the immunized groups with the experimental inactivating agents were compared with the ones from traditional HCHO. Cut-off value was defined as mean OD of diluent control +2 SD. Results as shown in (Figure 3) revealed that for anti-PV1, H_2_O_2_ response was non-significantly different from HCHO response (*p* = 0.6611). Although AA showed a clear elevated response compared to HCHO, it was not statistically different (*p* = 0.0678) while EGCG (which is not drastically different from AA) showed superiority to HCHO statistically (*p* = 0.0147). Anti-PV2 OD values for AA, EGCG and H_2_O_2_ were all non-significantly different from HCHO (*p* = 0.2627, 0.2851 and 0.7916 respectively). Concerning anti-PV3, both AA and EGCG showed elevated OD responses than HCHO (*p* = 0.0459 and 0.0216 respectively) whereas H_2_O_2_ readings were non-significantly different from HCHO (*p* = 0.4562). As for negative control groups, their OD readings were below cut-off values for the three types.

### 3.7. Polioviruses-Neutralizing Antibodies

Anti-polio neutralizing antibodies in rats’ sera were tested by their ability to neutralize a challenging dose of polioviruses. Challenge viruses type 1, 2 and 3 were found to have the titers of 75, 56 and 56 TCID50/0.05 mL respectively which are within the accepted ranges (50–200 TCID_50_/0.05 mL). Results represented in (Figure 4) showed that for HCHO, geometric mean titers of anti-polio neutralizing antibodies types 1, 2 and 3 were 24, 21 and 32 respectively. H_2_O_2_ results were 30, 18 and 39 while AA titers were 29, 22 and 36. Finally, EGCG results were 29, 23 and 35. The observed results indicated different responses; however, the statistical analysis showed that all the results of our experimental inactivating agents were non-significantly different from traditional HCHO.

## 4. Discussion

The mass production of affordable Sabin strain based inactivated polio vaccine (sIPV) for routine immunization is considered a milestone in poliomyelitis eradication plan especially in low- and middle-income countries. In the context of this global goal, a numerous number of researches were conducted to provide safe, effective and affordable sIPV. This study investigated different alternatives (H_2_O_2_, AA and EGCG) for formaldehyde as an inactivating agent in sIPV production. All these inactivating agents were chosen in our study on the bases of their previously proved safety, antiviral activity and cost-effectiveness.

This study demonstrated the ability of H_2_O_2_, AA and EGCG to achieve a complete and irreversible inactivation of poliomyelitis Sabin strains type1, 2 and 3. The inactivation was assured using Hep-2 cells which are known for their sensitivity to polio viruses and officially used for titration of Sabin poliomyelitis vaccine [29]. The virus suspensions were subjected to our experimental inactivating agents as well as formaldehyde till the viruses were no longer detectible using titration test. The results showed that H_2_O_2_, AA and EGCG were able to inactivate polioviruses in less than 24 h while HCHO required 4 days of inactivation. Based on historical data that formaldehyde inactivation curves follow a nonlinear regression model [35], this model was applied to all our inactivation curves. The obtained results suggested that the experimental inactivating agents are significantly less time-consuming than HCHO which can ultimately affect the production time hence the cost of production.

Due to the criticality of the inactivation step in IPV production process; testing for the presence of residual active polioviruses after inactivation process is a necessity and is considered as a crucial test for vaccine safety. Although WHO recommended a minimum of 1500 human doses to be tested for effectiveness of inactivation; this volume depends on the D-antigen content of the prepared vaccines and was not available for our research-level study [5]. Other than that; all other recommendations were followed including that the dilution of the inoculated samples did not exceed 1:4 in the nutrient medium also the area of the cells monolayer in each flask per milliliter of inoculum = 8 cm^2^ (<3 cm^2^/1 mL inoculum). It is quite difficult to detect virus residuals and they require a much longer time to produce a CPE than the untreated viruses [29]. For that; the test was performed in original cultures followed by two successive subcultures and these subcultures were observed for the maximum period of time as technically possible (approximately three weeks) to allow any traces of active virus to replicate and produce a visible CPE. This test confirmed the complete inactivation of all the treated viral suspensions.

While rapid and complete inactivation is considered as an important characteristic of an inactivating agent; preservation of the antigenic epitopes of the virus is also an important objective. Previous studies raised concerns about the reduction of potency in inactivated Sabin polio vaccines as formaldehyde caused damaging of the antigenic epitopes of polioviruses [36]. Levels of Polioviruses D-antigen epitopes remained undamaged after the different inactivation processes were assessed in this study by ELISA. Results revealed the superiority of H_2_O_2_, AA and EGCG in preserving significantly higher levels of viral antigenic moieties than traditional HCHO.

Previous studies showed antigenic differences between traditional Salk-IPV and IPV made from Sabin strains [37]. These studies recommended the formulation of a new international reference standard to measure the D- antigen content and assess the potency of the s-IPV products [38]. For this reason; four identical trivalent vaccine preparations were formulated; the only difference between them is the inactivating agent and these preparations were tested for their safety and immunogenicity. Concerning the safety of the preparations; it was confirmed via test for the effectiveness of inactivation which was performed on each inactivated monovalent bulk and discussed previously besides abnormal toxicity test which is a general unspecific safety test for vaccines for human use where trivalent vaccine preparations were injected in mice and guinea pigs and the test was conducted as per European pharmacopoeia monograph.

Another critical parameter for the quality control of IPV is the evaluation of the potency or immunogenicity of the vaccine. D-antigen content is a reliable simple quantitative assay to determine IPV potency and it is used as a routine quality control test for batch release to reduce time and cost of testing and to align with the global direction to reduce the usage of experimental animals [39]. However, In-vivo rat serology assay simulates the response of a biological system against administrated antigens and it is a direct evaluation of the vaccine effectiveness [40].

Immunogenicity of our vaccine preparations was investigated via their ability to produce anti-polio specific antibodies in Wistar rats. The rats’ sera were tested for the presence of antibodies against polio via two tests; ELISA assay and micro-neutralization assay in tissue culture micro-plates system. ELISA results reviled the comparability and even superiority of some types of the experimental preparations over the traditional formaldehyde vaccine preparation. Previous studies demonstrated the adverse effect of formaldehyde on the antigenicity and immunogenicity of IPV and suggested the usage of other means for inactivation such as beta-propiolactone [28] or even gamma radiation [5]. Despite the fact that ELISA can provide a good indication for the immune response; poliovirus neutralizing antibodies test is the reference test chosen by the WHO to reflect the level of protection and the immunity of polio vaccine preparations [41]. This test investigates the ability of the produced serum antibodies to specifically neutralize a certain dose of poliovirus. Previous studies showed that neutralizing antibodies level should be above 1:8 to be considered protective [42]. All our preparations produced protective levels of antibodies against polioviruses and were comparable to traditional formaldehyde.

Based on the shown superiority in the antigen preservation of our experimental inactivating agents; a similar superiority was anticipated in the induction of antibodies in rats. However, the obtained immune response results from this experiment did not reflect that. This could be partially explained by the yet to be cleared correlation between in vitro and in vivo potency assays in case of sIPV products. Such correlation has been long established and proved for conventional Salk IPV vaccines [38]. Also, previous studies showed that in some cases the loss of the D-antigen did not impact significantly on the in vivo potency of sIPV vaccine [40]. Another explanation could be the use of “statistical significance” itself in the interpretation of results. In the last few years, many calls for omitting the use of P value and statistical significance to accept or reject the research hypothesis were made [43,44]. The statistically non- significant differences do not mean that the data are not different; they rather indicate that many factors such as the sample size and the number of replicates can affect the experiment and consequently might give different results [45].

Therefore, further investigation will be carried out using different doses of the preparations and analyzing the antibodies levels at different time intervals for better understanding of the immune responses of our preparations and exploring the possibility of antigen sparing per human dose which will definitely cause more reduction in the cost of the vaccine.

Another future approach to be investigated is the use of novel oral polio vaccine strains (nOPVs) for the production of sIPV. Novel OPVs are genetically modified versions from Sabin OPV strains designed to be more stable and less likely to regain virulence causing cVDPV cases. In the last few years, a novel OPV type-2 vaccine (nOPV-2) has been developed as a part of the strategies for the response to cVDPV-2s, adopted by GPEI [46]. Two vaccine candidates of nOPV-2 are currently in clinical trials [47] while nOPV-1 and nOPV-3 are in preclinical stage [48]. Novel OPV strains are now investigated to be used as critical tools in cases of outbreaks to achieve and maintain eradication of polio. However, they might also serve as an alternative for standard Sabin strains in the production of sIPV. This approach can minimize the biohazard risk imposed by the polio vaccine manufacturing facilities in the post-eradication era. Our experimental inactivating agents which demonstrated their ability to inactivate traditional Sabine polioviruses may have the potential to be effective on the modified strains as well.

## 5. Conclusions

To the best of our knowledge, this is the first controlled study to investigate H_2_O_2_, AA and EGCG inactivation abilities on Sabin polioviruses. We can conclude that these inactivating agents have a potential for being used as potent, safe and time- saving alternatives for formaldehyde.

## Figures and Tables

**Figure 1 vaccines-08-00601-f001:**
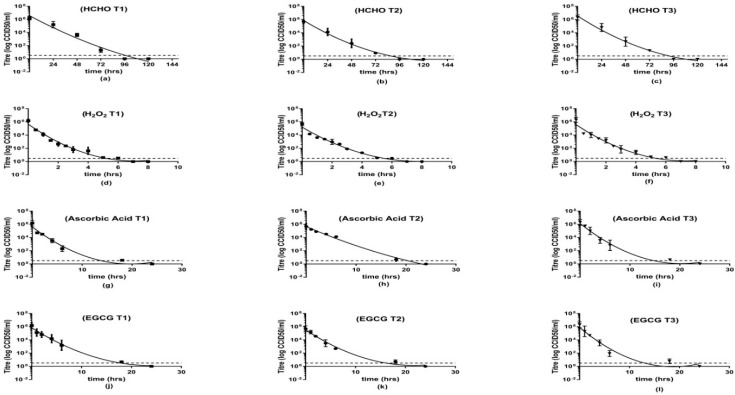
Inactivation kinetics of the three PV serotypes using different inactivating agents. Viral infectivity titers of the aliquots were determined by titration using Hep-2c cells in tissue culture microtiter plates. Each point is the average of two independent tests to measure the virus titer after a specific exposure time to the inactivating agent tested using tissue culture technique and expressed as log_10_CCID_50_/_mL_ and the error bars show the standard deviation (SD). The curves are fitted using non linear regression of the data. PV1 is represented as squares, PV2 as circles and PV3 as triangles. The method limit of detection (0.5 log CCID_50_/_mL_) is indicated by horizontal dashed line also symbols below LOD indicate no detected CPE at those time points. (**a**–**c**) Effect of HCHO on the infectivity titers of PV 1, 2 and 3 respectively; (**d**–**f**) Effect of H_2_O_2_ on the infectivity titers of PV 1, 2 and 3 respectively; (**g**–**i**) Effect of ascorbic acid on the infectivity titers of PV 1, 2 and 3 respectively and (**j**–**l**) Effect of EGCG on the infectivity titers of PV 1, 2 and 3 respectively.

**Figure 2 vaccines-08-00601-f002:**
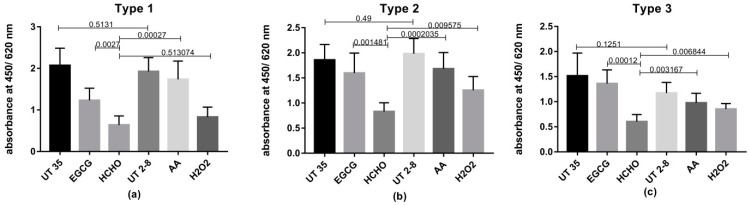
Retention of D-antigens of the three PV serotypes following inactivation by different inactivating agents determined using ELISA. Each column represents the mean absorbance values obtained from the different inactivated vaccine preparations when tested at the end of inactivation period. Two untreated control groups of polioviruses one kept at 35 °C (UT35) and the other at 2–8 °C (UT2-8) were tested as well. The error bars represent the standard deviation (SD). The capped lines represent the compared groups and the numbers written above the capped lines are the *p*-values. (**a**) PV1 results, (**b**) PV2 results and (**c**) PV3 results.

**Figure 3 vaccines-08-00601-f003:**
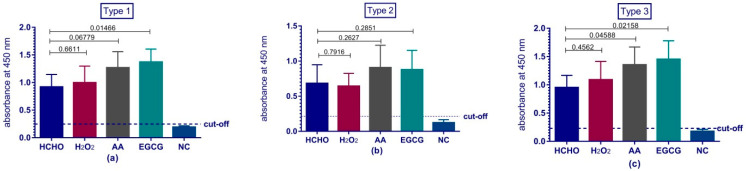
Immune response to the inactivated trivalent bulks in rats’ sera IgG antibodies were determined by ELISA. Serial two fold dilutions of sera obtained from rats immunized with one of the four vaccine preparations (HCHO, H_2_O_2_, AA or EGCG) and the sera of unimmunized control group (NC). Data is the mean absorbance of duplicate wells and error bars show the standard deviation (SD). Cut-off value is mean absorbance of diluent control +2 SD and is represented by a horizontal dashed line. The capped lines represent the compared groups and the numbers written above the capped lines are the *p*-values. (**a**) PV1 results, (**b**) PV2 results and (**c**) PV3 results.

**Figure 4 vaccines-08-00601-f004:**
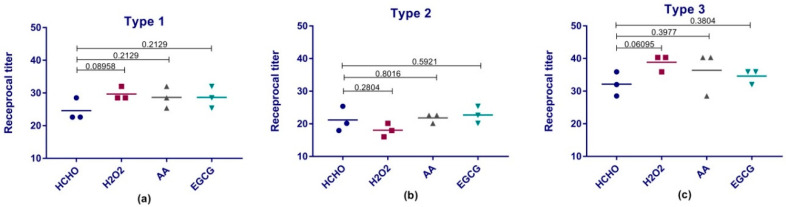
Sabin viruses neutralizing antibodies against PV serotypes. Titers of neutralizing antibodies obtained from each group of rat sera after immunization with one of the four vaccine preparations were tested via tissue culture technique. Titers are expressed as reciprocals. The test was performed in triplicate for each serotype and the geometric mean titers (GMT) are represented by horizontal lines. The capped lines represent the compared groups and the numbers written above the capped lines are the *p*-values. (**a**) PV1, (**b**) PV2 and (**c**) PV3.

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
