# Peer review of "In-Vitro Inactivation of Sabin-Polioviruses for Development of Safe and Effective Polio Vaccine"

_vaccines, 2020, doi:10.3390/vaccines8040601_

Round 1

Reviewer 1 Report

The authors have addressed all the points raised by the reviewers.

Reviewer 2 Report

No additional suggestions or comments. The Revised manuscript is well written and the response to reviewers' comments appropriate and clear.

Reviewer 3 Report

The authors' response to the reviewer's comments is acceptable. 

This manuscript is a resubmission of an earlier submission. The following is a list of the peer review reports and author responses from that submission.

Round 1

Reviewer 1 Report

In this work, the authors compared several protocols that could be used to inactivate poliovirus strains for the purpose of producing an inactivated vaccine. The effectiveness of the protocols was evaluated by checking the residual infectivity after inactivation. The antigenic properties of the inactivated virus suspensions were also evaluated through different techniques. Mainly, this study was properly conducted and addresses a topical subject, ie the improvement of the current polio vaccines. Please find my comments and suggestions below:

  • My main concern is about the use of “statistical significance”. Line 212, the authors indicate that “p-values < 0.05 were considered statistically significant”. Although this statement is common in biomedical publications, its relevance is questioned (read for example https://www.nature.com/articles/d41586-019-00857-9). I think it would be preferable to indicate the p-values and let the readers decide whether or not the observed differences are significant or not. For instance, in Figure 3, EGCG is considered as significantly higher than the control while AA is not. Yet, AA and EGCG do not differ drastically one from the other. Line 267, the authors indicate that AGCG “showed superiority to HCOH (p=0.0147)”, which is true, but the superiority is very very tiny! Similarly, line 286, the statement that “all the results… were non-significantly different from traditional HCHO” is true but it does not mean that there is no differences: rather, it means that the number of replicates is too low to determine whether or not the differences that are observed are significant or not.
  • Could the authors indicate the poliovirus strains used for the ELISA and the seroneutralization assays?
  • Line 43. I do not think that OPV-3 will be withdrawn because of WPV3 eradication. Do the authors have references that support this idea?

Author Response

Response to Reviewer 1 Comments

Point 1: My main concern is about the use of “statistical significance”. Line 212, the authors indicate that “p-values < 0.05 were considered statistically significant”. Although this statement is common in biomedical publications, its relevance is questioned (read for example https://www.nature.com/articles/d41586-019-00857-9). I think it would be preferable to indicate the p-values and let the readers decide whether or not the observed differences are significant or not. For instance, in Figure 3, EGCG is considered as significantly higher than the control while AA is not. Yet, AA and EGCG do not differ drastically one from the other. Line 267, the authors indicate that AGCG “showed superiority to HCOH (p=0.0147)”, which is true, but the superiority is very very tiny! Similarly, line 286, the statement that “all the results… were non-significantly different from traditional HCHO” is true but it does not mean that there is no differences: rather, it means that the number of replicates is too low to determine whether or not the differences that are observed are significant or not.

Response 1: The authors would like to thank the reviewer for raising such important and valuable comment. Unfortunately, many statisticians call for waiving the concept of “statistical significance” when it comes to the interpretation of results. The argument about the relevance of the statistical significance to the actual meaning of the results is legitimate, however; this is still the most commonly used method to describe and compare the results in the vast majority of scientific publications. Some scientists suggest the use of alternative statistical tools such as Bayesian analysis but they concluded that the biggest issue is not in the use of “statistical significance” rather than the misinterpretation of the phrase itself in the scientific culture.

Although “not stating the statistical significance” is a progressive and revolutionary approach, it may confuse the readers and make it harder to draw a proper conclusion from the study. Besides, it is still not a commonly used approach by most academic editors and reviewers.

Our final conclusion did not claim “superiority” of the experimental inactivating methods over formaldehyde in terms of antigenicity or immunogenicity. We only claimed “comparability” and that they have the potential of being used as alternatives on the bases of their safety and time-saving which is consequently reflected on the cost. Therefore, the conclusion did not misuse the statistical significance.   

Editing:

Results:

Rephrase the results to clarify any discrepancies and indicate the observed differences along with the statistical differences.

  • Lines 262 to 264:

Rephrase “H2O2 recorded significantly higher results for PV2 and PV3 (P= 0.0096 and 0.0068), however; it showed insignificant difference from HCHO result for PV1 (P=0.5131).” to be “H2O2 recorded significantly higher results for PV2 and PV3 (P= 0.0096 and 0.0068). Although the observed results of H2O2- inactivated PV1 are higher than HCHO results, this increase was insignificant statistically (P=0.5131).”

  • Lines 282 to 285:

Rephrase “Results as shown in (Figure 3) revealed that for anti-PV1, AA and H2O2 responses were non-significantly different from HCHO response (P= 0.0678 and 0.6611 respectively) while EGCG showed superiority to HCHO (P=0.0147).” to be “Results as shown in (Figure 3) revealed that for anti-PV1, H2O2 response was non-significantly different from HCHO response (P= 0.6611). Although AA showed a clear elevated response compared to HCHO, it was not statistically different (P=0.0678) while EGCG (which is not drastically different from AA) showed superiority to HCHO statistically (P=0.0147).

  • Lines 305 to 307:

Rephrase “All the results of our experimental inactivating agents were non-significantly different from traditional HCHO.” To be “The observed results indicated different responses; however, the statistical analysis showed that all the results of our experimental inactivating agents were non-significantly different from traditional HCHO.”    

Discussion:

Addition of a paragraph in the discussion section concerning the misconception of the statistical significance and elaborating that the experiments if repeated may give different results depending on the number of replicates, sample size and many other factors.

  • Lines 387 to 392:

Add” Another explanation could be the use of “statistical significance” itself in the interpretation of results. In the last few years, many calls for omitting the use of P value and statistical significance to accept or reject the research hypothesis were made [43, 44].  The statistically non- significant differences do not mean that the data are not different; they rather indicate that many factors such as the sample size and the number of replicates can affect the experiment and consequently might give different results [45].”

Point 2: Could the authors indicate the poliovirus strains used for the ELISA and the seroneutralization assays?

Response 2: The poliovirus strains used in both tests are the same monovalent bulks that were used to prepare the vaccine preparations Type 1 (LSc, 2ab), Type 2 (P 712, Ch, 2ab) and Type 3 (Leon 12a, 1b) which are described in details in point (2.2 Virus strains).

Editing:

  • Lines 189- 190:

Rephrase “coated with100µl/well polioviruses serotypes (1, 2 or 3)” to be “coated with 100µl/well Sabin polioviruses monovalent bulks (GSK, Belgium) types 1, 2 or 3”

  • Line 214-215:

Rephrase “After that, polioviruses types 1, 2 and 3 were diluted” to be “After that, Sabine polioviruses monovalent bulks (GSK, Belgium) types 1, 2 and 3 were diluted”

Point 3: Line 43. I do not think that OPV-3 will be withdrawn because of WPV3 eradication. Do the authors have references that support this idea?

Response 3: The following references mentioned the possibility of OPV-3 withdrawal.

  • Potential Future Use, Costs, and Value of Poliovirus Vaccines

https://doi.org/10.1111/risa.13557

“Moreover, the Global Certification Commission recently certified the eradication of WPV3 (World Health Organization, 2019a), which creates the possibility of globally coordinated serotype 3 OPV (OPV3) cessation prior to serotype 1 OPV (OPV1) cessation. “

  • A novel gamma radiation-inactivated sabin-based polio vaccine

doi: 10.1371/journal.pone.0228006

“With the eradication of wild type PV2, the trivalent OPV was replaced by bivalent OPV (bOPV; types 1 and 3) [7], lacking Sabin-2; and a similar strategy is expected for Sabin-3 now that wild type PV3 has been declared eradicated”

Editing:

  • Line 45:

Citation for these two references added at the end of the sentence.

Reviewer 2 Report

Manufacture of inactivated polio vaccine (IPV) Involves growth of large amounts of virulent Salk strains of poliovirus which needs time to inactivate. One method for reducing risk of breaches of virulent virus during manufacture is to inactivate attenuated Sabin oral poliovirus strains (sIPV) as a substitute for the Salk strains.  The authors demonstrate successful alternative methods to inactivate monovalent bulk preparations of the three Sabin poliovirus serotypes. The methods were less expensive than gold standard inactivation by formaldehyde; The time needed for inactivation was also shorter; and the sIPV was antigenic.

The paper was well written and is of interest for the global polio eradication program. However, there are two points that still need to be addressed.

.1. Required: The authors indicate that significant high D-antigen levels were observed using their alternative production methods. The authors correctly reference the paper by Crawt et al Differences in Antigenic Structure of Inactivated Polio Vaccines Made From Sabin Live-Attenuated and Wild-Type Poliovirus Strains: Impact on Vaccine Potency Assays that demonstrated that there were differences in the reactivity of antibody reagents to classical IPV and sIPV products.

The description of the antibodies used to test this in the current manuscript in the materials and methods section 2.6 (lines 135-153) is not sufficiently detailed to document whether or not D-antigens were measured. Please provide catalog numbers for the polyclonal rabbit anti-polio antibodies that were obtained from Sanofi Pasteur and used to test the D-antigen content, indicate what was used as antigen for the  production of the antibodies,  and indicate whether the antibodies are formally recommended for use in measuring D-antigens of IPV or sIPV. If these antibodies have not been specifically designated to measure D-antigens then the authors should obtain a standard sIPV D-antigen preparation (for example WHO IS 17/160,NIBSC) and include results of a standard curve test to demonstrate that the antibodies used, measure D-antigen content of  sIPV. They could also consider a negative control test for D antigen content, poliovirus heated at 56  degree C to inactivate the D-antigen content.

.2. Safety. It is true that growing large stocks of attenuated Sabin strains instead of neurovirulent Salk strains for IPV production is safer for production of IPV. However, Sabin strains can still cause poliomyelitis. Moreover, according the WHO Global Action Plan III (GAP III), Manufacturing processes with type 2 Sabin and shortly type 3 Sabin 3 strains must be done in “polio essential facilities” requiring high levels of biosecurity and biosafety and annual certification. Please add a paragraph in the discussion relating to novel OPVs (genetically modified for stability and safety) that are currently in clinical trials but probably will not need to be produced in “polio essential facilities.  These nOPVs will be shortly released for emergency use to combat outbreaks of vaccine derived type 2 polioviruses. The authors might suggest that their alternative inactivation methods which they demonstrate work on standard Sabin strains can and should be applied to producing sIPVs from these nOPV strains.

.Query: How easy is it to scale up the inactivation methods from laboratory to the large bulk amounts of virus  that are needed for vaccine production lines?  

Author Response

Response to Reviewer 2 Comments

Point 1: The authors indicate that significant high D-antigen levels were observed using their alternative production methods. The authors correctly reference the paper by Crawt et al Differences in Antigenic Structure of Inactivated Polio Vaccines Made From Sabin Live-Attenuated and Wild-Type Poliovirus Strains: Impact on Vaccine Potency Assays that demonstrated that there were differences in the reactivity of antibody reagents to classical IPV and sIPV products.

The description of the antibodies used to test this in the current manuscript in the materials and methods section 2.6 (lines 135-153) is not sufficiently detailed to document whether or not D-antigens were measured. Please provide catalog numbers for the polyclonal rabbit anti-polio antibodies that were obtained from Sanofi Pasteur and used to test the D-antigen content, indicate what was used as antigen for the production of the antibodies, and indicate whether the antibodies are formally recommended for use in measuring D-antigens of IPV or sIPV. If these antibodies have not been specifically designated to measure D-antigens then the authors should obtain a standard sIPV D-antigen preparation (for example WHO IS 17/160,NIBSC) and include results of a standard curve test to demonstrate that the antibodies used, measure D-antigen content of  sIPV. They could also consider a negative control test for D antigen content, poliovirus heated at 56 degree C to inactivate the D-antigen content.

Response 1: The authors would like to thank the reviewer for his positive feedback and important comment

The used procedure for polio D- antigen titration by ELISA is the standard in-house developed procedure used during the routine quality control batch release testing of IPV vaccines produced by Sanofi Pasteur either standalone vaccine (Imovax) or the combined vaccines (Hexaxim) and the standard operating procedure (SOP) is fully validated by Sanofi.

The heifer and rabbit anti-polio antibodies are also in-house developed, tested and used specifically in polio D-antigen content test for Sanofi products. They were obtained from Sanofi as part of the analysis requirements to assess the potency of each IPV batch before releasing it in the Egyptian market. 

These antibodies are not commercially available; therefore, catalog numbers are not applicable. They have only code numbers which are used internally.

The immunization strains used to prepare the antibodies are as following:

Type 1: Mahoney- type 1

Type 2: MEF1- type 2

Type 3: Saukett- type 3

Editing:

Materials and methods

  • Lines 84-92:

Addition of a new heading (2.3 Antibodies) and description of the antibodies. 

Point 2: Safety. It is true that growing large stocks of attenuated Sabin strains instead of neurovirulent Salk strains for IPV production is safer for production of IPV. However, Sabin strains can still cause poliomyelitis. Moreover, according the WHO Global Action Plan III (GAP III), Manufacturing processes with type 2 Sabin and shortly type 3 Sabin 3 strains must be done in “polio essential facilities” requiring high levels of biosecurity and biosafety and annual certification. Please add a paragraph in the discussion relating to novel OPVs (genetically modified for stability and safety) that are currently in clinical trials but probably will not need to be produced in “polio essential facilities.  These nOPVs will be shortly released for emergency use to combat outbreaks of vaccine derived type 2 polioviruses. The authors might suggest that their alternative inactivation methods which they demonstrate work on standard Sabin strains can and should be applied to producing sIPVs from these nOPV strains.

Response 2: Novel OPVs are still in their early stages of development. The promising results of phase 1 clinical trials on nOPV-2 vaccine candidates strongly suggest its use during outbreaks instead of standard OPV.

The idea of using these modified strains to produce sIPV is actually a promising approach to be pursued.

Editing:

Discussion:

  • Lines 397- 407:

Addition of a new paragraph in the discussion section related to nOPV.

Point 3: Query: How easy is it to scale up the inactivation methods from laboratory to the large bulk amounts of virus that are needed for vaccine production lines? 

Response 3: Any critical change in vaccine production requires a series of procedures. First of all, characterization of the inactivated viruses and validation for the effectiveness of inactivation process on both pilot and production scales. Then, full process validation, batch analysis and stability testing. Pre-clinical studies are required and clinical data may be required to assure that the change did not affect the quality and safety of the vaccine

Reviewer 3 Report

Abd-Elghaffar, A.A et al., elegantly wrote the manuscript about the alternative approaches to inactivate the Sabin-polioviruses for the development of a safe and effective polio vaccine.  To achieve this objective PV strains have been individually treated with H2O2, ascorbic acid, and EGCG, and those were compared with the classic inactivating agent formaldehyde. The authors' main concern was to avoid the time-consuming process as well as thermal degradation and destruction of epitopes exposed to elevated temperatures for a longer period of time by using formaldehyde.  Provided data support that the experimental inactivating agents consumed less-time, 24 hours,  than the conventional HCHO, and retained the of D-antigens to some extent.  Even though the elicitation of immune response and the production of neutralizing antibodies by those experimental inactivating agents were insignificant compared to the classical inactivating agent HCHO which are important parameters to be considered for a vaccine.  With the given data authors claim that these agents are safer and cost-effective than the conventional agent.   

Major concerns:

  1. Authors repeatedly claiming about the cost-effectiveness of alternative experimental inactivating agents, but reviewer doesn’t find that comparisons in the form of a table or graph. Above all as we are in the pandemic situation, the cost will not be the factor when it comes to the production of the vaccine, therefore, cost-effectiveness can be described differently or alternatively removed from the manuscript.

Minor corrections:

  1. P values can be inserted onto the graph itself instead of on the body of the results for figure 2,3,4,5.
  2. Line 298, Formaldehyde, “f” can be lower case.
  3. The supplier's complete address can be included.
  4. Line 212 can be clarified, is that 0.05 is considered as insignificant or significant?

Author Response

Response to Reviewer 3 Comments

Point 1: Major concerns: Authors repeatedly claiming about the cost-effectiveness of alternative experimental inactivating agents, but reviewer doesn’t find that comparisons in the form of a table or graph. Above all as we are in the pandemic situation, the cost will not be the factor when it comes to the production of the vaccine, therefore, cost-effectiveness can be described differently or alternatively removed from the manuscript.

Response 1: The authors would like to thank the reviewer for his insightful comments

In a pandemic situation, cost is not the concern for vaccine production, which is entirely true in cases such as COVID- 19, Ebola and other pandemic crises. However, the case of IPV production is different.

Since 2008, the WHO started a call for affordable IPV options especially for low and middle- income countries. In order to enforce the use of IPV universally, the price/immunization dose has to be lowered and mass production of IPV is required to meet the global demand. Different approaches were investigated to achieve this goal.

During the last four years (2016-2020) after the global switch from tOPV to bOPV and cessation of OPV-2, many countries (including Egypt) failed to fulfill their commitment on the prescheduled timeline to introduce a single dose of IPV in the routine immunization due to the high cost and the global shortage. All of this can emphasize the importance of cost effectiveness in case of IPV.   

Unfortunately, authors are not able to predict the cost of sIPV production using the experimental inactivating agents compared with HCHO. However, the cost-effectiveness rises from different factors:

Reduction of inactivation time affects the time and consequently the cost of production.

All the proposed inactivating agents are safe during handling and storage.

The bio-hazardous and environmental risks emerge from handling of large quantities of formaldehyde ultimately increase the cost of production.   

Editing:

Abstract:

  • Line: 30

Replace “cost –effective” with “ time- saving” 

Discussion:

  • Line 332

Add “which can ultimately affect the production time hence the cost of production” to clarify the main source of cost- effectiveness.

Conclusion:

  • Line: 411

Remove “cost- effective”

Point 2: Minor corrections:

P values can be inserted onto the graph itself instead of on the body of the results for figure 2,3,4,5.

Response 2:  Editing: Figures 2,3 and 4:

Addition of the P-values on the graph.

Addition of “The capped lines represent the compared groups and the numbers written above the capped lines are the P- values.” In legends of the figures.

NB: the manuscript does not contain figure 5.

Point 3: Line 298, Formaldehyde, “f” can be lower case.

Response 3: Editing Line 320: Typographical error corrected.

Point 4: The supplier's complete address can be included.

Response 4: Editing: Addition of the manufacturing countries for all the suppliers in the materials and methods section.

Point 5: Line 212 can be clarified, is that 0.05 is considered as insignificant or significant?

Response 5: Editing Line 227: Addition of “(typically ≤ 0.05)” for clarification.